# Enhanced Stability and In Vitro Biocompatibility of Chitosan-Coated Lipid Vesicles for Indomethacin Delivery

**DOI:** 10.3390/pharmaceutics16121574

**Published:** 2024-12-09

**Authors:** Angy Abu Koush, Eliza Gratiela Popa, Daniela Angelica Pricop, Loredana Nita, Cezar-Ilie Foia, Ana-Maria Raluca Pauna, Beatrice Rozalina Buca, Liliana Lacramioara Pavel, Liliana Mititelu-Tartau

**Affiliations:** 1Department of Pharmacology, Faculty of Medicine, ‘Grigore T. Popa’ University of Medicine and Pharmacy, 700115 Iasi, Romania; maierean_angy@yahoo.com (A.A.K.); cezar030699@gmail.com (C.-I.F.); beatrice-rozalina.buca@umfiasi.ro (B.R.B.); liliana.tartau@umfiasi.ro (L.M.-T.); 2Department of Pharmaceutical Technology, Faculty of Pharmacy, ‘Grigore T. Popa’ University of Medicine and Pharmacy, 700115 Iasi, Romania; 3Research Center with Integrated Techniques for Atmospheric Aerosol Investigation in Romania, RECENT AIR, Laboratory of Astronomy and Astrophysics, Astronomical Observatory, Department of Physics, ‘Al. I. Cuza’ University, 700506 Iasi, Romania; daniela.a.pricop@gmail.com; 4‘P. Poni’ Institute of Macromolecular Chemistry of Romanian Academy, 700487 Iasi, Romania; lnazare@icmpp.ro; 5Department of Anatomy, Faculty of Medicine, ‘Grigore T. Popa’ University of Medicine and Pharmacy, 700115 Iasi, Romania; paunaanamariaraluca@gmail.com; 6Department of Morphological and Functional Sciences, Faculty of Medicine and Pharmacy, ‘Dunarea de Jos’ University, 800010 Galati, Romania; doctorpavel2012@yahoo.com

**Keywords:** indomethacin, chitosan, lipid vesicles, biocompatibility, mice

## Abstract

Background: Lipid vesicles, especially those utilizing biocompatible materials like chitosan (CHIT), hold significant promise for enhancing the stability and release characteristics of drugs such as indomethacin (IND), effectively overcoming the drawbacks associated with conventional drug formulations. Objectives: This study seeks to develop and characterize novel lipid vesicles composed of phosphatidylcholine and CHIT that encapsulate indomethacin (IND-ves), as well as to evaluate their in vitro hemocompatibility. Methods: The systems encapsulating IND were prepared using a molecular droplet self-assembly technique, involving the dissolution of lipids, cholesterol, and indomethacin in ethanol, followed by sonication and the gradual incorporation of a CHIT solution to form stable vesicular structures. The vesicles were characterized in terms of size, morphology, Zeta potential, and encapsulation efficiency and the profile release of drug was assessd. In vitro hemocompatibility was evaluated by measuring erythrocyte lysis and quantifying hemolysis rates. Results: The IND-ves exhibited an entrapment efficiency of 85%, with vesicles averaging 317.6 nm in size, and a Zeta potential of 24 mV, indicating good stability in suspension. In vitro release kinetics demonstrated an extended release profile of IND from the vesicles over 8 h, contrasting with the immediate release observed from plain drug solutions. The hemocompatibility assessment revealed that IND-ves exhibited minimal hemolysis, comparable to control groups, indicating good compatibility with erythrocytes. Conclusions: IND-ves provide a promising approach for modified indomethacin delivery, enhancing stability and hemocompatibility. These findings suggest their potential for effective NSAID delivery, with further in vivo studies required to explore clinical applications.

## 1. Introduction

A modern strategy for achieving effective drug delivery focuses on designing carrier systems that are informed by a detailed understanding of their interactions with the biological environment. This includes considering the target cell population, cell surface receptors, disease-induced receptor changes, and the active substance’s action mechanism, site of action, pharmacokinetics, molecular mechanisms, and disease pathobiology [1,2,3]. In recent decades, the development of sophisticated drug delivery systems has emerged as a major focus of pharmaceutical research, driven by the need to overcome challenges in drug bioavailability and therapeutic effectiveness [4,5].

Polymeric systems made from biodegradable and biocompatible natural or synthetic polymers represent a promising formulation for controlled drug delivery [6]. These polymers act as carriers to facilitate targeted delivery and control the release of active ingredients. Compared to conventional formulations, polymer-based systems can enhance drug solubility, which boosts absorption rates and reduces the necessary dosage for therapeutic effectiveness [7]. Additionally, these nanoparticles are noted for their stability and safety profile.

Modern drug delivery approaches increasingly aim to achieve targeted release of active compounds to extend their effects in the targeted area, minimizing adverse reactions [8,9]. However, creating systems for targeted delivery poses challenges in the development of agents with promising regional therapeutic effects, particularly in conditions involving inflammation and pain [10,11].

Non-steroidal anti-inflammatory drugs (NSAIDs) are widely used analgesic and anti-inflammatory agents that act primarily by inhibiting cyclooxygenase (COX) enzymes [12]. COX are enzymes that play a key role in the conversion of arachidonic acid into prostaglandins, which are lipid compounds involved in inflammation, pain, and other physiological processes. Cyclooxygenase-1 (COX-1) is an isoform constitutively expressed in most tissues and is involved in maintaining normal physiological functions, such as protecting the stomach lining, supporting renal function, and regulating platelet aggregation. Cyclooxygenase-2 (COX-2) is inducible and is primarily expressed in response to inflammatory stimuli, such as injury or infection. It is associated with the production of prostaglandins, which mediate inflammation, pain, and fever [12]. COX-1 inhibition, however, is associated with risks of gastrointestinal bleeding and peptic ulcers, whereas COX-2 inhibition carries potential cardiovascular risks. As a result, prolonged NSAID use can lead to complications such as peptic ulcers, gastrointestinal bleeding, and both cardiovascular and renal damage [13].

While current literature provides insights into the encapsulation of certain NSAIDs within nanoparticle-based systems, there is relatively limited information on their pharmacodynamic effects, especially in animal models of pain and inflammation. Previous studies have demonstrated that these advanced delivery systems can achieve efficient drug loading and prolonged release profiles under in vitro conditions [14,15]. However, findings regarding their in vivo pharmacodynamic effects in laboratory animals have been inconsistent.

Indomethacin (IND), a member of the NSAID family, provides anti-inflammatory, analgesic, and antipyretic effects by reducing prostaglandin production through cyclooxygenase inhibition, with a stronger inhibition of COX-1 than COX-2. Its analgesic and anti-inflammatory effects are attributed to blocking the biosynthesis and release of prostaglandins, which otherwise sensitize nociceptive C fibers to pain stimuli. This inhibition also reduces pain transmission triggered by substances like bradykinin, tumor necrosis factor-alpha, interleukins, and other pain-inducing molecules [12]. IND exhibits linear pharmacokinetics and is quickly absorbed from the gastrointestinal tract, achieving nearly 100% bioavailability after oral administration. As a weak organic acid, it binds extensively to plasma proteins (90%) and readily crosses the blood–brain barrier and placenta [16]. Its plasma elimination is biphasic, with a half-life of about 1 h in the initial phase and 2.6–11.2 h in the secondary phase. Given its high bioavailability, IND is minimally affected by hepatic first-pass metabolism, undergoing liver metabolism through glucuronidation, O-demethylation, and N-deacylation. The primary metabolites, O-desmethyl-indomethacin, O-deschlorobenzoyl-indomethacin, O-desmethyl-N-deschlorobenzoyl-indomethacin, and their glucuronide conjugates lack anti-inflammatory activity. IND and its metabolites are eliminated mainly via urine (60%) and, to a lesser extent, in bile and feces (34.5%) [17].

The common gastrointestinal issues associated with the use of IND, such as irritation, ulcers, and bleeding, occur due to the drug’s non-selective inhibition of COX enzymes [12]. The goal of controlled release systems is to prolong the therapeutic effect of the drug while minimizing peak plasma concentrations, which are often responsible for side effects like gastrointestinal irritation. By reducing the frequency of high doses and ensuring steady drug release, controlled delivery systems can reduce adverse effects and improve patient adherence.

Considering IND’s adverse gastrointestinal effects, incorporating it into nanosystems offers several pharmacological benefits: this approach could reduce gastrointestinal irritation and enhance anti-inflammatory and analgesic efficacy in experimental models. The lipid vesicles, particularly when coated with biocompatible polymers like chitosan, provided an effective solution to these challenges. The vesicles protect the drug from rapid degradation, control its release over time, and help target the drug to specific sites, reducing the impact on the gastrointestinal tract. Additionally, the chitosan coating enhanced stability and further helped to reduce irritation in the gastric mucosa, offering a dual benefit of improved drug delivery and reduced side effects. Although CHIT is widely recognized for its biocompatibility, stability, and ability to enhance mucosal penetration, studies applying it specifically to IND encapsulation are sparse. Indomethacin, often regarded as a gold standard among NSAIDs for its unmatched anti-inflammatory potency, remains an enigma in modern therapeutics, its potential being overshadowed by a legacy of significant side effects, limiting its widespread use and study. Most existing research focuses on other NSAIDs such as ibuprofen [18], or diclofenac [19], leaving a gap in understanding IND’s interaction with CHIT and its potential therapeutic advantages. Poly(lactic-co-glycolic acid) (PLGA)-based systems are well-established for IND delivery due to their biodegradable and sustained-release properties. However, they often require surfactants like polyvinyl alcohol for stabilization, which can pose toxicity concerns [20]. CHIT offers a natural alternative with a better safety profile and does not rely on such additives, reducing the risk of adverse reactions. While several systems (e.g., liposomes and PLGA nanoparticles) aim to mitigate NSAID-induced gastrointestinal damage, CHIT-coated vesicles uniquely combine the protective effects of encapsulation with CHIT’s known ability to enhance mucosal barrier function [19]. Moreover, the electrostatic stabilization provided by CHIT-coated vesicles enhances their stability [21]. This dual action addresses a critical gap in achieving both efficacy and safety in IND delivery.

This study aimed to design, characterize, and evaluate the biocompatibility of novel IND-ves in mice.

## 2. Materials and Methods

### 2.1. Substances

IND (catalog code: I7378, purity 98.5%, molecular weight: 357.79 g/mol) and the materials needed for nanoparticle preparation, i.e., CHIT (sourced from crab shells, catalog code: C3646), phosphatidylcholine (catalog code: P5638, derived from soy, type II-S, containing 14–29% choline), cholesterol (catalog code: C8667, ≥99% purity, molecular weight: 386.65 g/mol), chloroform (catalog code: C2432, stabilized with 100–200 ppm amylene, ≥99.5% purity, molecular weight: 119.38 g/mol), glacial acetic acid (catalog code: 695092, ≥99.7% purity, molecular weight: 60.05 g/mol), and ethyl alcohol (catalog code: E7148, 95.0% purity, molecular weight: 46.07 g/mol), were supplied by Sigma-Aldrich Chemical Co., Steinheim, Germany (www.sigma-aldrich.com, accessed on 15 October 2024).

The CHIT used in the preparation featured an 80% degree of N-deacetylation, an average molecular weight (Mw) of 310,000 g/mol, and a polydispersity index of 3.26. A 0.25% (*w*/*w*) CHIT solution was prepared using a 0.5% (*v*/*v*) acetic acid solution.

### 2.2. Animals

The study involved healthy, genetically unmodified, three-month-old male Swiss mice (25–30 g), sourced from the “Cantacuzino” National Medical-Military Institute for Research and Development, Băneasa Station, Bucharest, Romania, through the biobase of the “Grigore T. Popa” University of Medicine and Pharmacy, Iași, located at the Advanced Center for Research and Development in Experimental Medicine (CEMEX).

The mice were housed individually in Plexiglas cages under controlled conditions, with a stable temperature of 21 ± 2 °C, relative humidity of 50–70%, and a 12-h light/dark cycle. To mitigate chronobiological effects, the tests were conducted between 8:00 and 12:00 a.m. The animals received standardized pelleted diet, with daily monitoring of individual consumption.

Each experimental group consisted of five mice that received the test substances, while a separate control group, was given distilled water and served for comparative analysis. The substances were administered as a single dose via intragastric gavage according to the following protocol: Group 1 (Control): 0.1 mL distilled water/10 g body weight; Group 2 (CHIT): 0.1 mL of CHIT solution/10 g body weight; Group 3 (IND): IND solution 5 mg/kg body weight; Group 4 (IND-ves): CHIT-based vesicles entrapping IND 5 mg/kg body weight.

### 2.3. Preparation Technique of IND-Loaded Lipid Vesicles

To formulate lipid vesicles loaded with IND, a molecular droplet self-assembly technique was employed for encapsulating the hydrophobic compound. Ethanol was used as a solvent for the lipids, drug, and cholesterol, while CHIT was dissolved in glacial acetic acid at a 0.5% concentration.

In this preparation, 0.09 g of soy lipid were dissolved in 0.66 mL of ethanol, while 0.015 g of cholesterol and 0.01 g of IND were each dissolved separately in 1 mL of ethanol. The combined lipid and IND solutions yielded a final volume of 1.66 mL, which was then subjected to ultrasound treatment at 25% amplitude for 10 min at 29 °C, generating 20,000 kJ of energy, using a Bandelin SONOPULS 2450 ultrasonic homogenizer (Sigma-Aldrich, Steinheim, Germany). This sonication process disrupts multilamellar structures, converting them into unilamellar vesicles, having the size regulated by the amplitude of the ultrasound [22,23]. Following sonication, the resulting solution was gradually injected into 8.33 mL of double-distilled water maintained at 22–23 °C. For this process, double-deionized water, obtained using a Liston A1104 distillation unit (Biosan, Zhukov, Russian Federation), was utilized. The resulting IND-loaded lipid vesicle suspension, referred to as IND-vl, appeared slightly translucent.

Separately, an equivalent ethanol mixture volume (1.66 mL) was added to 8.33 mL of 0.25% CHIT solution under magnetic stirring at 800 rpm for 20 min at 22 °C, producing IND-encapsulating lipid vesicles, designated as IND-ves. Both dispersions underwent an additional 5-min stirring. Adding CHIT to the drug-loaded vesicles induced significant changes in their size and morphology, enhancing the stability of the colloidal system.

The pH value of both dispersions was measured using a Sartorius Professional PP-50 pH meter (Sartorius Lab Instruments GmbH & Co. KG, Göttingen, Germany). To remove residual lipids and any unencapsulated drug and to bring the pH close to physiological levels, the dispersion was dialyzed for 2 h. Dialysis for both CHIT-coated and non-CHIT-coated suspensions was conducted using tubular fiber membranes with a 12,000 Da MWCO pore size (catalog code D6191-25EA), supplied by Sigma-Aldrich Chemical Co. (Steinheim, Germany). Following dialysis, physiological pH values were achieved, with a pH of 7.0 for the IND suspension and 6.7 for the IND-ves dispersion.

### 2.4. Characterization of IND-Loaded Vesicle Dispersions

#### 2.4.1. Morphological Analysis, Size, and Zeta Potential Measurement

The hydrodynamic size and stability of the IND-loaded vesicle dispersions were determined using a Zetasizer Nano ZS ZEN-3500 (Malvern Instruments, Worcestershire, UK). Dark-field micrographs were obtained using an Olympus BX61 optical microscope (CytoViva, Boston, MA, USA) with 10× and 40× oil immersion objectives.

For structural characterization, scanning electron microscopy (SEM) was conducted to obtain micrographs of IND-ves using the EDAX-Quanta 200 system (Thermo Fisher Scientific, Eindhoven, The Netherlands). SEM was employed in this study to investigate the surface morphology and size distribution of the lipid vesicles. SEM provides high-resolution imaging of external features, enabling detailed visualization of vesicle shape, surface texture, and structural integrity. These parameters are critical for characterizing the stability and physical properties of the nanoformulation, which are directly related to its potential efficacy and biocompatibility. By focusing on SEM, we aimed to ensure comprehensive characterization of the vesicle surfaces, which are integral to understanding their drug delivery potential [24]. The average diameter of over 200 vesicles was measured from SEM images using ImageJ software (version 1.8.0). Samples were prepared following a strict protocol. Each suspension was diluted with distilled water, and 0.1 mL of each suspension was placed on specialized grids designed for SEM. The suspensions were centrifuged with a Hettich Microliter centrifuge (Hettich Holding GmbH & Co., Kirchlengern, Germany) at 15,000 rpm.

UV-Vis spectra were recorded on a Shimadzu Pharma Spec 1700 UV-Vis spectrophotometer (ALT, San Diego, CA, USA) with 1 cm path-length quartz cells, and FTIR (Fourier Transform Infrared) spectra were acquired with a Vertex 70 spectrophotometer (Bruker, Karlsruhe, Germany).

#### 2.4.2. Evaluation of IND Encapsulation Efficiency in Vesicles

To assess the drug encapsulation efficiency and measure the release rate of IND, the calibration curve for IND dissolved in ethanol was generated. Initially, IND was dissolved in ethanol to prepare a concentrated stock solution, which was subsequently diluted to create six distinct concentrations.

For evaluating the encapsulation efficiency of IND in vesicles, both with and without a CHIT coating, 3 mL aliquots from each suspension was collected. These aliquots were centrifuged at 15,000 rpm for 60 min to separate the supernatant from the pellet; the supernatant was removed, and was replaced with an equivalent volume of ethanol.

The pellets, supplemented with ethanol, from both coated and uncoated dispersions, were then subjected to sonication for 120 min to disrupt the vesicle membranes. This process was conducted after a 4-h incubation at a constant temperature. For spectrophotometric analysis, 0.1 mL of the sonicated suspension was mixed with 0.9 mL of water and 2 mL of ethanol in a quartz cuvette. Both samples, analyzed in the spectral range of 200–400 nm showed a peak absorbance at 320 nm, indicative of IND.

The encapsulation efficiency (EE%) of IND was calculated using the following formula: EE% = [(Ai − Ar) × 100]/Ai, where Ai is the initial amount of IND and Ar represents the amount of IND released from the vesicles.

### 2.5. In Vitro Studies

#### 2.5.1. In Vitro Release Profile Assessment

The in vitro release kinetics of IND from IND-ves was assessed over a period of 10 h using a dialysis separation method, which enabled the determination of the drug’s molar concentration at various time points. A volume of 10 mL dispersion of IND-ves was placed in a cellulose acetate dialysis bag (MW 12–14 KDa, Sigma Aldrich, Hamburg, Germany). This bag was submerged in a 200 mL container filled with phosphate-buffered saline at pH 7.4 (Sigma Aldrich Chemical Co., Steinheim, Germany). The release medium was continuously stirred magnetically at 100 rpm while maintained at a temperature of 37 ± 0.5 °C.

The molar concentration of IND was measured spectrophotometrically using a Shimadzu Pharma Spec 1700 UV-Vis spectrophotometer (ALT, San Diego, CA, USA). Absorbance readings were taken at 320 nm by withdrawing 2 mL of the release medium at specified intervals: 15 min, 30 min, 45 min, 60 min, 90 min, 1 h, 3 h, 4 h, 6 h, and 8 h. After each measurement, an equal volume of the dissolution medium was replaced in the assay compartment. The concentration of IND in each sample was calculated based on the established calibration curve, allowing for quantitative analysis of the IND release from the vesicles and forming the foundation for the release kinetics study.

The dissolution test for the IND solution was conducted by preparing a stock solution of IND dissolved in ethanol, which was then diluted to the desired concentration. The solution was placed in a suitable dissolution medium, such as phosphate-buffered saline or another appropriate buffer, at a constant temperature (typically 37 °C) under stirring conditions. The dissolution process was monitored over a defined period, and samples were taken at specified time intervals. The released IND from the solution was quantified using the same spectrophotometric method employed for the IND-ves dispersion. This allowed for a direct comparison of the release kinetics between the free drug solution and the encapsulated drug in the vesicles. The absorbance of the solution was measured at 320 nm, the characteristic absorption peak for IND, and used to calculate the amount of drug released at each time point. The release rate (R) was defined as the number of moles of IND released per unit time per unit volume of the release medium: R(t) = 1/V × dν(t), where V represents the volume of the medium and dν signifies the number of moles of IND released in the time interval dt. The molar concentration at a given time (t) was expressed by the equation: C(t) = ν(t)/V, where C is the molar concentration of IND released into the medium and ν is the number of moles of IND at time t within the volume V.

#### 2.5.2. In Vitro Hemocompatibility Evaluation

To evaluate the hemocompatibility of the tested substances, we employed an in vitro method that assessed their impact on erythrocyte architecture and viability by quantifying hemoglobin release into the plasma due to erythrocyte lysis induced by contact with potentially toxic agents [25,26].

Mice were locally anesthetized using 1% benzocaine to minimize discomfort and facilitate efficient blood collection. Blood samples were collected from the lateral tail vein into heparinized vacutainers to prevent coagulation and preserve erythrocyte viability. The samples were then centrifuged at 5 °C for 5 min at a relative centrifugal force (RCF) of 1500 to separate the erythrocytes from the plasma.

The isolated erythrocytes were washed three times with saline to remove residual plasma and minimize the influence of plasma components on subsequent measurements. A 2% (*v*/*v*) erythrocyte suspension was prepared by diluting the erythrocytes in saline, followed by incubation at 37 °C for 45 min. This suspension served as the negative control (Neg-c) for the experiments.

Triton X-100, known for its hemolytic properties, was used at a concentration of 10% (*v*/*v*) and incubated for 45 min at 37 °C, serving as the positive control (Poz-c) for hemolysis evaluation. The erythrocyte suspension was then exposed to the test substances for 45 min at 37 °C to allow direct interaction with the erythrocyte membranes. After incubation, the suspension was centrifuged for 10 min at 1000 RCF to remove intact cells. The resulting supernatant, containing plasma and lysed erythrocytes, was critical for assessing hemoglobin release. The absorbance of the supernatant was measured at 540 nm, the specific wavelength for detecting released hemoglobin, using a Hewlett Packard 8453 UV-VIS spectrophotometer (Waldbronn, Germany).

The percentage of hemolysis (%) was calculated using the following formula: hemolysis (%) = (Absorbance of substance tested − Absorbance Neg-c) × 100/(Absorbance Poz-c − Absorbance Neg-c) [27].

This formula quantifies the hemolytic effect of the test substances by comparing them to the positive and negative controls, thus ensuring the validity of the results.

### 2.6. Statistical Processing of Data

The data were compiled and statistically analyzed using the WINDOWS EXCEL program, which facilitated the application of specific functions to calculate the statistical parameters necessary for characterizing the distribution series. Differences between the results obtained from the experimental groups and the control group were assessed using SPSS software, version 17.0 for Windows, employing the one-way ANOVA method for variance analysis.

The statistical analysis was further complemented by Tukey and Newman–Keuls post hoc tests for multiple comparisons, establishing the hierarchy of the impact intensity on the evaluated parameters for each test, particularly in the analysis of various biological findings from the laboratory investigations. A *p*-value of less than 0.05 or 0.01, compared to the control group, was considered statistically significant.

### 2.7. Ethical Aspects of the Research

The experimental research protocol received the ethical approval (Certificate No. 362/28 November 2023; DSV Project Authorization No. 69/15 January 2024). The study was conducted in accordance with the recommendations of the Ethics Commission of UMF “Grigore T. Popa” in Iași and aligned with international standards regarding the use of laboratory animals. The experiments adhered to national regulations and international directives concerning the protection of animals for scientific purposes, in accordance with the latest regulations on animal experimentation [28,29].

## 3. Results

### 3.1. pH Values

To coat the vesicles with CHIT, the dispersion needed adjustment to a pH close to 5 (Table 1). This was achieved by adding acetic acid, which helps prevent polymer clumping and promotes an even distribution of particles within the dispersion.

Initially, the CHIT lipid vesicles containing IND, in their undialyzed form, had an acidic pH of 5.00, which made them unsuitable for administration to laboratory animals due to potential irritation. In order to raise the pH value of these vesicles to a more physiological level, they were subjected to dialysis, resulting in a colloidal dispersion with a final pH of 6.7 (Table 1). This adjustment enhances the dispersion’s safety and suitability for administration, aligning better with the physiological requirements of the laboratory subjects.

### 3.2. Characteristics of Microvesicles

#### 3.2.1. Morphology

The morphological characterization of IND-vl and IND-ves dispersions was performed by closely examining dark-field optical microscopy images and SEM micrographs.

Dark-field imaging revealed distinct morphological differences in the self-assembly patterns of uncoated and CHIT-coated vesicles, with notable details visible at both 10× and 40× magnification. For IND-vl, a multilamellar self-assembly pattern was observed, featuring multiple overlapping lipid layers (Figure 1a).

In contrast, IND-ves displayed a more uniform and well-defined morphology (Figure 1b), with a consistent shape indicating enhanced stability. This stability is likely due to electrostatic interactions between CHIT and the lipid membranes.

The SEM images show that individualization of these vesicles could improve the delivery of active substances, as each vesicle is an individual unit, allowing for a more uniform distribution within the biological system. The images also revealed that IND-vl vesicles have a rougher surface and a greater degree of agglomeration, consistent with less controlled and more heterogeneous self-assembly. In contrast, the IND-ves vesicles displayed a smoother, more uniform surface, suggesting enhanced stability and greater resilience to external influences.

Using ImageJ software, a widely used tool for image analysis, the size distributions of the two vesicle types were measured from the SEM images, providing accurate particle size assessments and relevant statistical data. IND-vl samples showed an average diameter of 219.25 nm (Figure 2a), while IND-ves were larger, with an average diameter of 289.0 nm (Figure 2b).

#### 3.2.2. Hydrodynamic Dimensions 

The hydrodynamic sizes of IND-vl and IND-ves were analyzed by measuring vesicle counts across different sizes, providing detailed information on their distribution and uniformity in dispersion.

IND-vl showed an average hydrodynamic size of 274.5 nm, with a polydispersity index (PI) of 0.398 (Figure 3a), indicating a relatively uniform distribution of particle sizes. For IND-ves, the average hydrodynamic size was slightly larger at 317.6 nm, and the PI was slightly lower at 0.364 (Figure 3b).

#### 3.2.3. Zeta Potential

The Zeta potential analysis revealed that the IND-vl sample had a negative average Zeta potential of −19.2 mV (Figure 4a), indicating a predominantly negative surface charge on the uncoated lipid vesicles. In contrast, the IND-ves sample showed a positive Zeta potential of 24 mV (Figure 4b), suggesting a positive surface charge due to the CHIT coating on the vesicles.

#### 3.2.4. Infrared Absorption Spectra

Infrared absorption spectra were analyzed over a wavelength range of 4000 to 400 cm^−1^. The FTIR spectrum of the IND-vl sample revealed several characteristic bands representative of the vibrations of phospholipids, the primary components of vesicular membranes. The absorption band observed between 3451 and 3398 cm^−1^ corresponds to the stretching of –OH groups and intermolecular hydrogen bonds. The band at 1735 cm^−1^, which falls within the range of 1765 to 1720 cm^−1^, is attributed to C=O (carbonyl) stretching vibrations. The peak at 1236 cm^−1^ is associated with the P=O (phosphoryl) group. The absorption peak at 1173 cm^−1^, located within the range of 1200 to 1145 cm^−1^, corresponds to the stretching of the PO_2_ (phosphate) group. The peak at 1077 cm^−1^ is linked to the C–O–C stretching. The absorption bands at 970 cm^−1^ and 754 cm^−1^ are attributed to the vibrations of the P–O–C group (Figure 5).

The FTIR analysis offered valuable insights into the chemical structure and morphology of the IND-ves. The absorption band at 1600 cm^−1^ corresponds to the stretching vibrations of carbon-carbon (C=C) double bonds found in aromatic systems. The presence of this band indicates the integration of IND molecules into the lipid membrane, thereby enhancing its stability. The peak at 1468 cm^−1^ is attributed to the deformation of the O–CH_2_ bonds. The absorption point at 720 cm^−1^ is associated with the aromatic ring of IND. The peak at 1372 cm^−1^ corresponds to the symmetric deformation of the CH group, which is characteristic of cholesterol present in the lipid membrane (Figure 5).

#### 3.2.5. The Incorporation Efficiency of IND in IND-ves

The results from the spectrophotometric analysis indicated a marked difference in encapsulation efficiency between uncoated and CHIT-coated vesicles.

To assess the encapsulation efficiency of the drug, a calibration curve of IND dissolved in ethanol was created (Figure 6a). The IND-vl sample showed an encapsulation efficiency of approximately 71%, while the IND-ves sample achieved a notably higher encapsulation efficiency of 85% (Figure 6b). This finding suggests that the CHIT coating improves drug retention within the vesicular system.

#### 3.2.6. In Vitro Release Profile of IND from IND-ves

Dissolution tests revealed notable differences in the release profiles of the two formulations. Analysis of in vitro release kinetics indicated that IND-ves had a substantially (*p* < 0.001) slower IND release compared to the rapid release seen in the plain drug solution.

Within the first 30 min, 42.3% of IND was released from the plain solution, indicating rapid release, whereas only 1.4% was released from IND-ves, showing a delayed release. After 2 h, the plain solution had released 91.5% of IND, reaching 99.3% at 3 h, and achieving complete release by 4 h. In contrast, the release profile of IND from IND-ves showed that only 10.7% of the drug was released after 2 h (*p* < 0.001), increasing to 33.8% after 3 h (*p* < 0.001). It took until 7 h for 98.2% of the drug to be released from IND-ves (*p* < 0.001), with full release occurring at 8 h (Figure 7).

#### 3.2.7. In Vitro Hemocompatibility

Exposure to Triton X-100 resulted in a hemolysis rate of 84.45 ± 2.37%, which was statistically significant (** *p* < 0.001), compared to the negative control group that was not exposed to the detergent. In contrast, CHIT administration produced minimal hemolysis, with a rate of only 2.18 ± 0.04% (*p* > 0.05), comparable to the negative control group (Table 2).

Similarly, incubation of erythrocyte suspensions with IND did not lead to significant hemolysis, showing a value of 2.21 ± 0.05% (*p* > 0.05), which was in line with the negative control. Additionally, exposure of erythrocytes to IND-ves resulted in a low hemolysis rate of 2.25 ± 0.03% (*p* > 0.05), with no statistically significant difference from the negative control (Table 2).

## 4. Discussion

In a previous study, we successfully developed polymeric matrices using a copolymer made from poly(2-hydroxyethyl methacrylate-co-3,9-divinyl-2,4,8,10-tetraoxaspiro[5.5]-undecane) and poly(aspartic acid). This initial work provided valuable insights into the encapsulation efficiency, release kinetics, and biocompatibility of IND within these copolymeric systems in mice [30] and in rats [31], demonstrating their potential as effective platforms for modified drug delivery.

In this research, we began with the premise that encapsulating IND in lipid vesicles offers an alternative approach to modifying its pharmacokinetics. We prepared vegetal lipid vesicles loaded with IND using a dropwise technique and coated them with high molecular weight, highly deacetylated CHIT. Phospholipids have proven effective as drug carriers for both hydrophilic and hydrophobic substances [32]. In recent years, studies have shown that liposomal encapsulation can extend drug half-life and improve slow-release efficiency [33]. For hydrophilic drugs, transport typically involves intestinal wall absorption, transit through blood vessels to the liver via the portal vein, and then access to systemic circulation [34]. In contrast, hydrophobic drugs encapsulated in lipid chains enter the lymphatic system after penetrating the intestinal walls and eventually connect to the circulatory system through the thoracic duct, prolonging their vascular residence and enabling more sustained release [35,36,37].

The effectiveness of these delivery systems relies heavily on precise size control, which has been a focus of research to tailor vesicle dimensions to specific medical applications. Because lipid vesicles are sensitive to pH fluctuations, bile salts, and pancreatic lipase, polymer coatings have been explored for added protection. Using CHIT as a coating for vesicles has increased drug absorption by extending retention time and bioavailability compared to unencapsulated drugs [38,39,40]. CHIT’s stabilizing properties have shown to enhance gastric mucosal permeability and prolong the retention of absorbed drugs [41]. Additionally, the cationic nature of CHIT chains may help reduce lipid oxidation [42].

We characterized the IND-loaded microvesicles in terms of morphology and structure, assessed their stability in suspension, evaluated the drug’s efficiency incorporation, and examined both in vitro release and in vitro hemocompatibility.

Dark-field imaging effectively illuminates denser structures against a dark background, allowing for clearer visualization of dispersed particles. The examination of IND-vl revealed a multilamellar self-assembly pattern, resulting in irregular and non-uniform shapes that suggest a diverse size distribution. These irregular forms may indicate instability or variability during their formation, potentially affecting how these vesicles interact with their environment or other biochemical components. In contrast, IND-ves exhibited a more uniform and well-defined morphology, with a consistent shape indicative of enhanced stability, attributed to the electrostatic interactions between CHIT and the lipid membranes.

The IND-vl sample displayed irregular and heterogeneous shapes, deviating from the typical spherical morphology expected of lipid vesicles, as observed in SEM images. This morphology suggests a more complex and potentially unstable self-assembly process, where irregular shapes may result from uncontrolled interactions among the lipid components. Such heterogeneous forms can significantly influence the biological behavior of the vesicles, impacting their interactions with cells and other biomolecules.

In comparison, IND-ves demonstrated a spherical appearance and high dimensional uniformity. This morphology is associated with improved stability and enhanced delivery capabilities for active substances. Spherical vesicles are more likely to distribute evenly in solution, facilitating their penetration through biological membranes. The observed dimensional uniformity indicates that the CHIT coating process was effective, leading to a homogeneous structure essential for therapeutic applications.

The size distribution of both samples was assessed using ImageJ software, a popular tool for image analysis, by measuring the vesicle diameters from the SEM images. SEM offers high-resolution imaging to examine vesicle morphology, surface features, and size distribution, allowing for a detailed visual comparison between uncoated and CHIT-coated vesicles. It highlights structural changes, such as smoother surfaces and enhanced uniformity after coating, and helps identify potential defects or irregularities in vesicle structure that may impact drug release or stability. While IND-vl exhibited relatively smaller sizes, this variability reflects its previously noted heterogeneous morphology. The larger size of IND-ves is attributed to the CHIT coating, which adds an additional layer around the lipid vesicles. This increase in size may enhance dispersion stability and delivery efficiency, providing better protection against environmental degradation.

The measured PI of 0.398 indicates a moderate variability in particle sizes, suggesting a somewhat heterogeneous distribution among the IND-vl vesicles. This PI value reveals that the vesicles have a structure that is not entirely uniform, with particles of varying sizes. Such variability could potentially influence how these vesicles interact with other molecules or particles in their environment. In contrast, the reduction in PI for the IND-ves indicates that the addition of CHIT helps stabilize the particles, minimizing size variation and enhancing dispersion uniformity. The increase in hydrodynamic size observed with the addition of CHIT results from the formation of a composite layer due to electrostatic interactions between the positively charged CHIT chains and the negatively charged lipid membrane of the vesicles.

As a cationic polymer, CHIT electrostatically binds to the surface of the anionic vesicles, creating a protective layer [43]. This coating not only increases the overall size of the vesicles but also enhances their stability in dispersion, preventing aggregation and shielding them from external influences. The development of this composite layer contributes to a more stable structure, improving dimensional uniformity and allowing the vesicles to exhibit greater resistance to environmental factors.

Vesicle dispersion stability is a critical factor in evaluating the performance and efficiency of drug delivery systems. A key parameter for assessing the stability of colloidal dispersions, such as lipid vesicle suspensions, is the Zeta potential, which measures the electric charge of the dispersion and provides insight into the electrostatic interactions between particles. It provides a rapid and non-destructive assessment of vesicle stability, making it essential for predicting the shelf-life of colloidal systems and ensuring their stability under physiological conditions.

To analyze the stability of the dispersions, two types of vesicular systems were compared: IND-vl and IND-ves. The negative Zeta potential observed for IND-vl indicates that the uncoated lipid vesicles carry a predominant negative electrical charge. This suggests that the electrostatic interactions between the particles are inadequate to prevent agglomeration. Typically, a Zeta potential around −20 mV is considered borderline and indicates moderate stability, which may imply a greater likelihood of particle aggregation that could reduce the efficiency of the drug delivery system.

In contrast, the positive Zeta potential of IND-ves corresponds to their positive electrical charge, providing a significant advantage in terms of stability. Particles with opposite electrical charges tend to repel one another, thus decreasing the probability of agglomeration.

FTIR analysis provides essential information regarding the chemical structure and morphology of the vesicles. By identifying the vibrational bands, it is possible to gain insights into the molecular interactions within vesicular systems and evaluate their stability. FTIR enables the non-invasive detection of drug–lipid interactions and polymer coatings without requiring extensive sample preparation.

For IND-vl, absorption bands show key functional groups contributing to membrane structure and stability. The presence of –OH stretching vibrations highlights hydrogen bonding with water, suggesting hydration at the polar head regions of phospholipids. The carbonyl (C=O) stretching band indicates that carbonyl groups within phospholipids help stabilize the membrane by reinforcing its structure. Phosphoryl (P=O) and phosphate (PO_2_) stretching vibrations underscore the role of polar groups in membrane stability and electrostatic interactions, affecting the vesicle’s surface charge and behavior in its environment. The C–O–C stretching indicates ether linkages essential for maintaining phospholipid cohesion, while P–O–C stretching suggests robust phosphate–lipid interactions that further support membrane integrity.

For IND-ves, specific absorption bands reveal IND’s integration into the lipid membrane, which enhances structural stability. The C=C stretching in aromatic rings indicates that IND molecules are embedded within the membrane, increasing rigidity and stability through van der Waals interactions with lipids. The O–CH_2_ deformation band highlights the interactions between IND’s functional groups and the lipid bilayer, signifying substantial interaction with membrane lipids. The aromatic ring band of IND confirms its incorporation, suggesting a firm fit within the lipid structure. Additionally, the CH deformation band suggests cholesterol’s presence, which is critical for membrane stability and influences interactions with IND molecules.

The FTIR spectrum of IND-ves closely resembles that of IND-vl, suggesting that the CHIT coating is physical rather than chemical in nature. This similarity indicates that no new peaks are present that would indicate the formation of covalent bonds between the CHIT chains and the lipids. These findings imply that the interactions between CHIT and the membrane lipids are primarily electrostatic and do not involve permanent chemical changes. As a result, while CHIT may enhance the stability of the vesicular system and improve the drug’s bioavailability, it does not create covalent bonds that alter the original structure of the phospholipids.

Encapsulation efficiency is an important indicator of a vesicular system’s ability to retain a drug, which is essential for developing pharmaceutical formulations that ensure adequate bioavailability of active substances. Spectrophotometric analysis revealed a significant difference in encapsulation efficiency between uncoated and CHIT-coated vesicles. For IND-vl, the encapsulation efficiency was around 71%, indicating that the remaining 29% was released into the surrounding medium. This loss could be attributed to various factors, such as the high solubility of IND in the medium, limited interactions between the drug and the lipid matrix of the vesicles, or the physical instability of the vesicular system. In contrast, the encapsulation efficiency of 85% in IND-ves indicates a substantial improvement in drug retention, which may lead to increased bioavailability and enhanced therapeutic efficacy. This improvement suggests that the electrostatic interactions between CHIT and the phospholipid membrane play a vital role in drug retention, thereby enhancing formulation stability.

The increase in encapsulation efficiency can be linked to the favorable electrostatic interactions between CHIT, which carries a positive charge, and the negatively charged lipid membranes [43]. These interactions help stabilize the vesicular structures, forming a matrix that retains IND more effectively. CHIT’s ability to engage electrostatically with membranes, which often possess a negative charge due to phospholipids, may create a stable complex that limits drug release. The CHIT coating likely functions as an additional barrier, controlling the diffusion rate of IND and enabling a more regulated and sustained release over time. Furthermore, it may enhance the physical stability of the vesicles, preventing aggregation and degradation. Improved stability ensures that the vesicular structure remains intact during storage and application.

The observed difference in encapsulation efficiency between the two vesicle types is statistically significant, indicating that the CHIT coating directly influences the delivery system’s performance. The improved encapsulation efficiency of IND-ves makes this formulation more promising for therapeutic applications, as enhanced drug retention within the vesicular system often leads to better clinical outcomes. The kinetic release profile of IND from the tested systems showed a rapid initial release, with approximately 42.3% released within the first half hour, reaching 99.3% after 3 h, and complete release occurring by 4 h. These results are consistent with existing literature, which indicates that NSAIDs, like IND, typically demonstrate a rapid dissolution profile for immediate therapeutic effect.

In contrast, the release of IND from IND-ves exhibited a more gradual pattern, starting with a latency period. After two hours, only 10.7% of IND was released, which increased to around 33.8% after 3 h, and reached 98.2% by 8 h, with total release achieved at this time. The release profile of IND-ves aligns with the trends observed in other biopolymer-coated systems, such as PLGA nanoparticles, which have reported extended release over 6–12 h [20]. This sustained release pattern suggests that the CHIT-stabilized vesicles effectively regulate the diffusion of IND, facilitating a modified release of the drug in the surrounding medium. The notable differences in release rates can be attributed to the encapsulation of IND molecules, which are evenly dispersed within the nanovesicles, with CHIT stabilization slowing their diffusion in the surrounding environment. CHIT coating provides a dual benefit of structural stability and controlled diffusion, outperforming uncoated lipid systems in sustained-release applications. The prolonged release of IND-ves can be attributed to the CHIT coating, which forms a semi-permeable barrier that slows drug diffusion. Additionally, electrostatic interactions between the positively charged CHIT and negatively charged IND molecules further regulate the release rate. The hydrophobic interactions between IND and the lipid bilayer core may also contribute to the observed latency in drug release. Combined with the stability conferred by pH adjustment and dialysis during formulation, these factors underline the superior performance of CHIT-coated vesicles in sustaining drug release.

Using a CHIT with a higher degree of deacetylation (D = 85%) and greater molecular weight likely results in the formation of more cationic charges from amino groups, saturating the lipid vesicle surface and increasing the system’s positive potential [43]. Additionally, the enhanced stability of the vesicles with CHIT coating was confirmed by the extended time (up to 4 h) needed to disintegrate the vesicle membrane under the influence of ethyl alcohol and ultrasonic fields. This modified release mechanism may be advantageous for applications requiring a gradual and sustained delivery of the drug, in contrast to the rapid action provided by conventional solutions.

Interaction between the erythrocyte suspension and the detergent Triton X-100 resulted in significant hemolysis, indicating considerable damage to the cell membranes and destruction of erythrocytes. However, incubating the erythrocyte suspension with CHIT did not result in noticeable hemolysis compared to the negative control group. Similarly, exposure of erythrocytes to IND and IND-ves caused minimal hemolysis, with no significant difference from the negative control, indicating good in vitro hemocompatibility. These results suggest that the tested substances do not adversely affect erythrocytes, demonstrating compatibility with them. Thus, the findings imply that IND-ves exhibits good in vitro hemocompatibility and does not significantly compromise erythrocyte integrity.

Research literature has highlighted a range of innovative strategies for developing IND-loaded nanoformulations, using either preformed polymers or lipid-based carriers [44,45]. Over time, scientists have explored various approaches to design nanosystems that optimize the transport and controlled release of IND. A variety of approaches have been explored for incorporating IND into nanoformulations to improve solubility, stability, and controlled release properties [45,46].

Researchers have developed various advanced systems to improve indomethacin (IND) delivery. Lipid-based nanocarriers, such as lipid emulsions, enhance IND’s solubility and bioavailability using phospholipids and surfactants [47,48]. Liposomal formulations, including sterically stabilized vesicles with stearylamines and cholesterol, have been designed to improve stability and sustain drug release [49,50]. Triacylglyceride-based conjugates with phosphatidylcholine and stearic acid extend release and enhance compatibility [51,52]. CHIT-based nanoparticles formed via ionotropic gelation offer controlled drug release [53].

Polymer-lipid nanoparticles combining lecithin and PLGA were created using double emulsion methods, providing stable, biocompatible systems [54,55]. Techniques like supercritical fluid processing with CO_2_ yielded PEG-stabilized nanoparticles with precise control over size and purity [56]. Electrohydrodynamic atomization produced uniform, monodisperse IND nanoparticles for targeted delivery [57]. Other studies developed amorphous nanosuspensions via wet milling with polyvinylpyrrolidone to maintain bioavailability [58] or used spray drying with cyclodextrins to enhance solubility and dissolution [59,60].

Biodegradable polymer systems were widely explored. Solvent evaporation or nanoprecipitation using PLGA or polycaprolactone matrices improved solubility, stability, and sustained release [61]. Scalable nanoprecipitation methods produced controlled-release nanoparticles [62]. Enteric-coated nanoparticles, made from materials like Eudragit L100, ensured intestinal release while protecting IND from gastric degradation [63]. Solid nanoparticles created with PLGA via solvent evaporation demonstrated biocompatibility and sustained release [64]. Lastly, microwave-assisted hydroxyapatite nanoparticles were developed for bone-targeted applications, incorporating IND within the crystal lattice or adsorbed on its surface [65].

These various nanoformulation approaches highlight the versatility of IND and the ongoing refinement of systems to optimize its therapeutic performance.

## 5. Conclusions

Phosphatidylcholine-based lipid microvesicles containing IND and stabilized with CHIT were successfully prepared using molecular droplet self-assembly. These vesicles exhibited uniform morphology, improved encapsulation efficiency, and enhanced stability, with the CHIT coating promoting a modified drug release profile. Compared to uncoated vesicles, the CHIT-coated vesicles showed slower release, which is beneficial for drugs requiring extended therapeutic effects. Microscopic analysis confirmed that CHIT-coated vesicles had a smooth, stable morphology, while uncoated vesicles were prone to aggregation. Hemocompatibility assays showed minimal hemolysis, indicating good in vitro compatibility. The positive Zeta potential of the CHIT-coated vesicles suggests potential for enhanced targeted delivery. Overall, these microvesicles offer a promising platform for drug delivery, with enhanced stability, controlled release, and good in vitro safety profile. Further studies should investigate the underlying mechanisms and in vivo efficacy of the CHIT-coated vesicles.

## Figures and Tables

**Figure 1 pharmaceutics-16-01574-f001:**
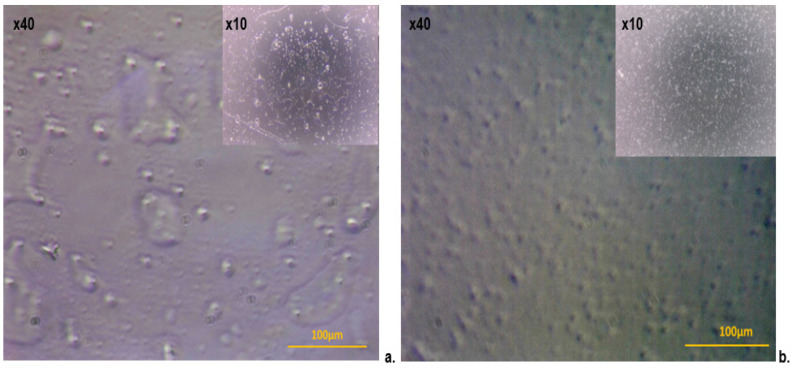
Dark-field optical microscopy images of IND-vl (**a**) and IND-ves (**b**), with 40× immersion objective (**a**,**b**); detail images were obtained with 10× objective.

**Figure 2 pharmaceutics-16-01574-f002:**
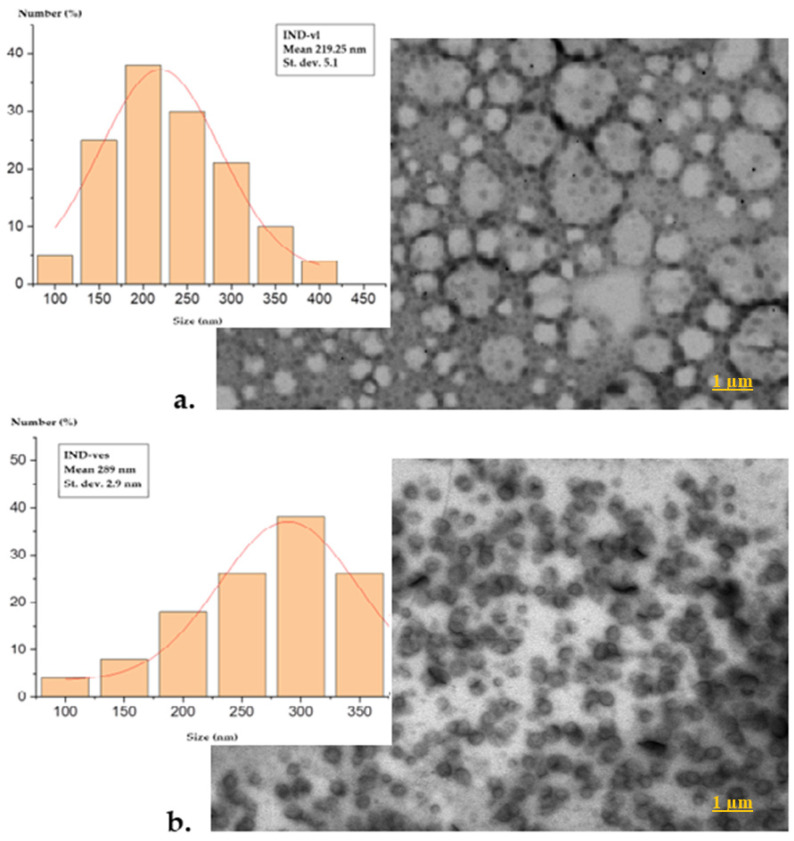
SEM microradiographs of IND-vl (**a**) and IND-ves (**b**).

**Figure 3 pharmaceutics-16-01574-f003:**
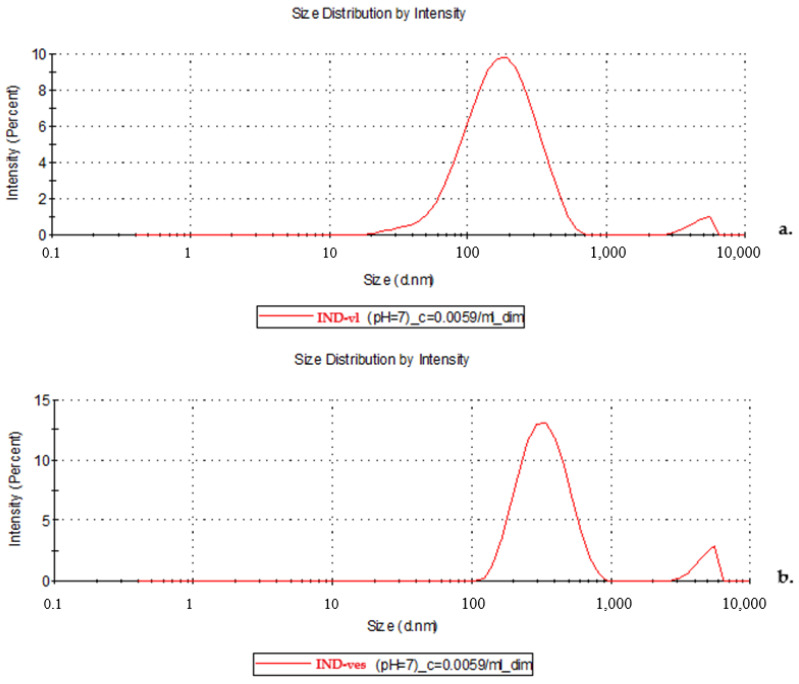
Hydrodynamic dimensions of IND-vl (**a**) and IND-ves (**b**).

**Figure 4 pharmaceutics-16-01574-f004:**
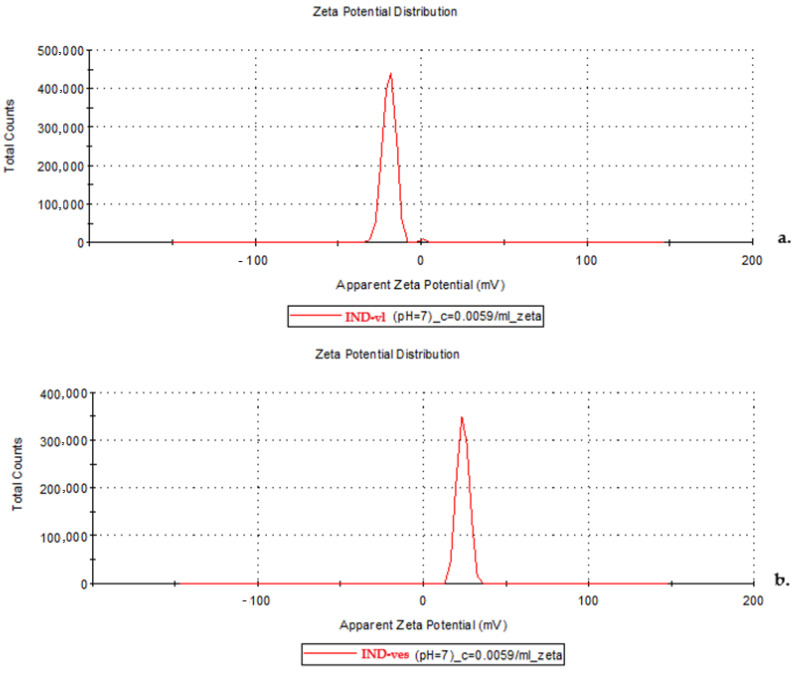
Zeta potential plot of IND-vl (**a**) and IND-ves (**b**).

**Figure 5 pharmaceutics-16-01574-f005:**
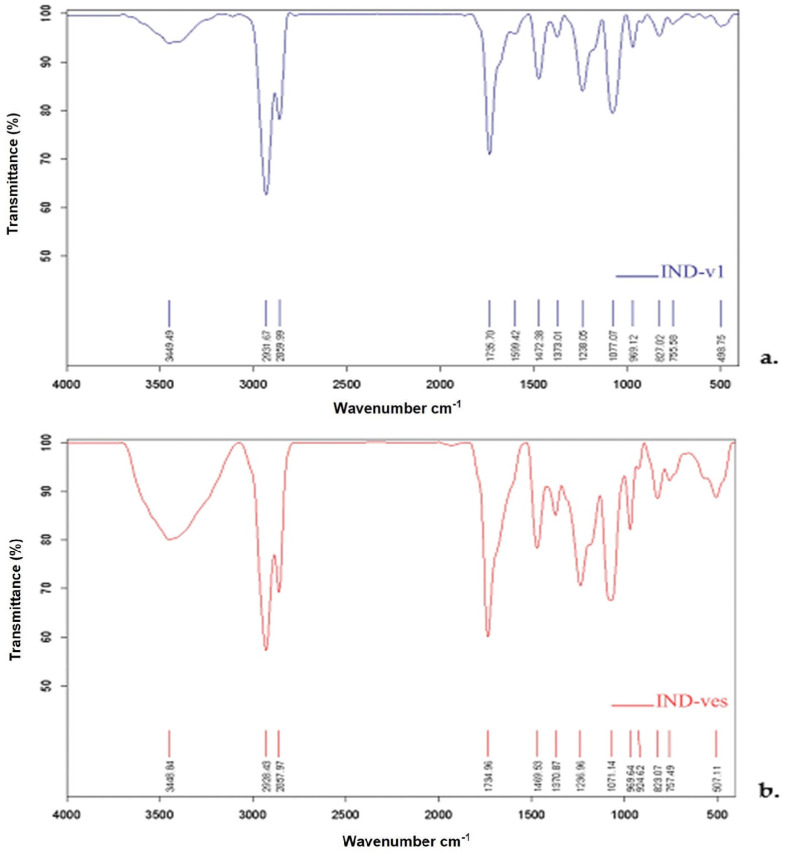
FTIR spectrum of IND-vl (**a**) and IND-ves (**b**).

**Figure 6 pharmaceutics-16-01574-f006:**
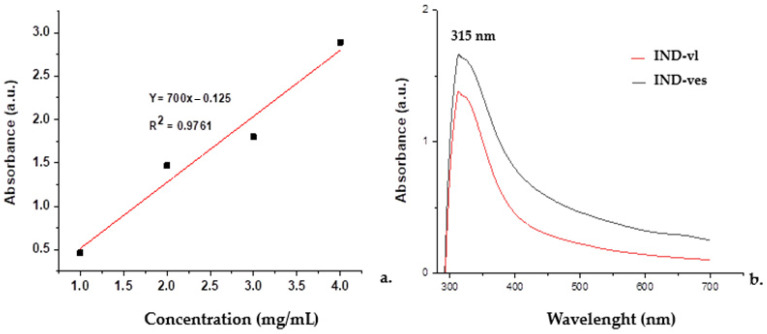
Calibration curve for IND (**a**) and the absorption spectra for IND-vl and IND-ves (**b**).

**Figure 7 pharmaceutics-16-01574-f007:**
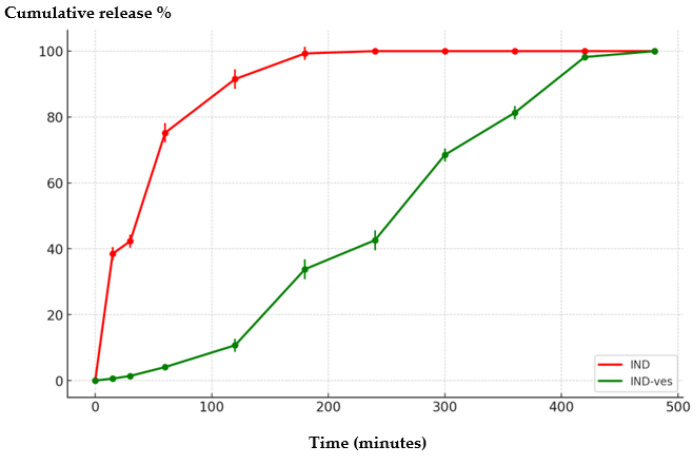
In vitro release curve of IND from IND-ves.

**Table 1 pharmaceutics-16-01574-t001:** The pH values of colloidal solutions with IND-vl and IND-ves.

Colloidal Solutions	pH
CHIT	7.35
Non-dialyzed IND-vl	5.50
Non-dialyzed IND-ves	5.00
Dialyzed IND-vl	7.00
Dialyzed IND-ves	6.70

Abbreviations: CHIT—chitosan, Non-dialyzed IND-vl—non-dialyzed lipid vesicles entrapping indomethacin, Non-dialyzed IND-ves—non-dialyzed lipid vesicles entrapping indomethacin coated with chitosan, Dialyzed IND-vl—dialyzed lipid vesicles entrapping indomethacin, Dialyzed IND-ves—non-dialyzed lipid vesicles entrapping indomethacin coated with chitosan.

**Table 2 pharmaceutics-16-01574-t002:** The in vitro hemocompatibility of IND-vl and IND-ves. Values are expressed as the arithmetic mean ± standard deviation (S.D.) for five animals per group. ** *p* < 0.001 compared to the negative control.

Group	Negative Control	Triton X-100	CHIT	IND-vl	IND-ves
% hemolysis	0.01 ± 0.01	84.45 ± 2.37 **	2.18 ± 0.04	2.21 ± 0.05	2.25 ± 0.03

## Data Availability

Data are contained within the article.

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
