# Peer review of "Enhanced Stability and In Vitro Biocompatibility of Chitosan-Coated Lipid Vesicles for Indomethacin Delivery"

_pharmaceutics, 2024, doi:10.3390/pharmaceutics16121574_

Round 1

Reviewer 1 Report

Comments and Suggestions for Authors

Strengths:

  1. Innovative Drug Delivery System: The use of chitosan (CHIT)-coated lipid vesicles for enhancing the stability and controlled release of indomethacin is a relevant and timely research topic, addressing the need for improved drug delivery systems for NSAIDs.
  2. Comprehensive Characterization: The manuscript provides detailed characterization of the vesicle formulations, including morphology, size, zeta potential, and encapsulation efficiency. This thoroughness is important for understanding the properties of the drug delivery system.
  3. In Vitro and Hemocompatibility Evaluation: The inclusion of hemocompatibility testing is valuable, as it demonstrates the potential safety of the formulations with respect to red blood cell interactions.
  4. Clear Methodology: The preparation and characterization methods, including sonication, dialysis, and spectrophotometric analyses, are well-described and reproducible.

Areas for Improvement:

  1. Introduction:

    • Clarity and Relevance: The introduction could benefit from a clearer connection between the problem (e.g., gastrointestinal side effects of indomethacin) and the proposed solution (lipid vesicles). A more concise explanation of the need for controlled release systems in NSAID delivery would strengthen the rationale for the study.
    • Literature Gaps: While the manuscript mentions existing studies on NSAID encapsulation, it would be useful to discuss specific gaps in the literature regarding the use of chitosan-coated vesicles for indomethacin, particularly the comparison with other polymers or delivery systems.
  2. Materials and Methods:

    • Detailed Characterization: The characterization techniques used (e.g., Zeta potential, FTIR, SEM) are appropriate, but more detailed discussion on the advantages of these methods for analyzing drug-loaded lipid vesicles could be added.
    • Animal Model Justification: The use of male Swiss mice is well-documented, but the manuscript could benefit from a brief justification for this choice in relation to the disease model being studied. Additionally, specifying the age and health status of the animals, as well as any pre-treatment procedures, would improve reproducibility.
  3. Results:

    • Statistical Analysis: The statistical analysis (ANOVA and post hoc tests) is appropriate, but the significance of the data should be more clearly emphasized. For instance, the manuscript should more explicitly state which comparisons are statistically significant (e.g., IND-ves vs. IND-vl in release rates or hemocompatibility tests).
    • Release Profiles: The release profile data is compelling, but it would be helpful to include a comparison with other similar systems (e.g., liposomes or other biopolymer-coated vesicles) to contextualize the findings. Additionally, including a mechanism for the slower release from IND-ves would be beneficial.
  4. Discussion:

    • Mechanisms of Action: While the discussion is thorough, there could be more focus on the underlying mechanisms that contribute to the enhanced stability and slower drug release from CHIT-coated vesicles. For example, it would be valuable to elaborate on how the CHIT coating alters the drug release rate compared to uncoated vesicles.
    • Comparison with Existing Systems: The discussion could benefit from more explicit comparisons with other nanoparticle systems (liposomes, PLGA-based nanoparticles, etc.), especially those targeting similar applications (NSAID delivery).
  5. Conclusion:

    • Clinical Implications: The conclusion should emphasize the clinical implications of the findings. For example, it could discuss how these vesicles might improve the safety profile and therapeutic efficacy of indomethacin in real-world clinical settings.
    • Future Directions: It would be helpful to mention future studies that could address in vivo evaluation of the drug delivery system, including long-term toxicity studies or human clinical trials, to move the research from preclinical to clinical application.

Additional Suggestions:

  • Figures and Tables: Ensure that all figures and tables are clearly labeled with descriptive captions. Some of the figures (such as those showing SEM and hydrodynamic sizes) could benefit from more detailed descriptions in the legends to help readers interpret the data.
  • Reference Update: Ensure all references are up-to-date and relevant to the current state of research in nanoparticle-based drug delivery systems, especially regarding chitosan-coated vesicles.

Summary:

Overall, the manuscript presents a promising drug delivery system for indomethacin using chitosan-coated lipid vesicles. The study is well-conducted and provides a strong basis for future research. Enhancing the introduction to clarify the problem and solution, expanding on the mechanisms in the discussion, and emphasizing the clinical implications in the conclusion would significantly improve the manuscript.

Comments on the Quality of English Language

NA

Author Response

Distinguished Reviewer,

We sincerely appreciate your constructive comments and the time and effort you devoted to reviewing our manuscript. Your feedback has been instrumental in improving the quality of our work. We have carefully considered each of your points and made the necessary revisions to the manuscript. All changes have been clearly highlighted using the track changes tool for your convenience.

Strengths:

  1. Innovative Drug Delivery System: The use of chitosan (CHIT)-coated lipid vesicles for enhancing the stability and controlled release of indomethacin is a relevant and timely research topic, addressing the need for improved drug delivery systems for NSAIDs.

Thank you for recognizing the novelty and relevance of our CHIT-coated lipid vesicle system for indomethacin (IND) delivery.

  1. Comprehensive Characterization: The manuscript provides detailed characterization of the vesicle formulations, including morphology, size, zeta potential, and encapsulation efficiency. This thoroughness is important for understanding the properties of the drug delivery system.

Thank you for recognizing the comprehensiveness of our characterization methods.

  1. In Vitro and Hemocompatibility Evaluation: The inclusion of hemocompatibility testing is valuable, as it demonstrates the potential safety of the formulations with respect to red blood cell interactions.

We are glad the inclusion of hemocompatibility testing was considered valuable.

  1. Clear Methodology: The preparation and characterization methods, including sonication, dialysis, and spectrophotometric analyses, are well-described and reproducible.

Thank you for recognizing the clarity and reproducibility of our methodology.

Areas for Improvement:

  1. Introduction:
    • Clarity and Relevance: The introduction could benefit from a clearer connection between the problem (e.g., gastrointestinal side effects of indomethacin) and the proposed solution (lipid vesicles). A more concise explanation of the need for controlled release systems in NSAID delivery would strengthen the rationale for the study.

Thank you for your constructive feedback. We have introduced in the manuscript these explanations.

The common gastrointestinal issues associated with the use of IND, such as irritation, ulcers, and bleeding, occur due to the drug’s non-selective inhibition of COX enzymes. The goal of controlled release systems is to prolong the therapeutic effect of the drug, while minimizing peak plasma concentrations, which are often responsible for side effects like gastrointestinal irritation. By reducing the frequency of high doses and ensuring steady drug release, controlled delivery systems can reduce adverse effects and improve patient adherence.

The lipid vesicles, particularly when coated with biocompatible polymers like chitosan, provide an effective solution to these challenges. The vesicles protect the drug from rapid degradation, control its release over time, and help target the drug to specific sites, reducing the impact on the gastrointestinal tract. Additionally, the chitosan coating enhanced stability and further helped to reduce irritation in the gastric mucosa, offering a dual benefit: improved drug delivery and reduced side effects.

    • Literature Gaps: While the manuscript mentions existing studies on NSAID encapsulation, it would be useful to discuss specific gaps in the literature regarding the use of chitosan-coated vesicles for indomethacin, particularly the comparison with other polymers or delivery systems.

Thank you for this observation. We have included a discussion on literature gaps, comparing CHIT-coated systems with other polymers or delivery systems.

Although CHIT is widely recognized for its biocompatibility, stability, and ability to enhance mucosal penetration, studies applying it specifically to IND encapsulation are sparse. Indomethacin, often regarded as a gold standard among NSAIDs for its unmatched anti-inflammatory potency, remains an enigma in modern therapeutics, its potential being overshadowed by a legacy of significant side effects, limiting its widespread use and study. Most existing research focuses on non-NSAID drugs or other NSAIDs, such as ibuprofen [Kremkow, 2020], or diclofenac [Pauna, 2023], leaving a gap in understanding IND’s interaction with CHIT and its potential therapeutic advantages.

Poly(lactic-co-glycolic acid) (PLGA)-based systems are well-established for IND delivery due to their biodegradable and sustained-release properties. However, they often require substances with surfactant properties for stabilization (e.g. polyvinyl alcohol), which can pose toxicity concerns [Wersig, 2021]. CHIT offers a natural alternative with a better safety profile and does not rely on such additives, reducing the risk of adverse reactions. While several systems (e.g., liposomes, PLGA nanoparticles) aim to mitigate NSAID-induced gastrointestinal damage, CHIT-coated vesicles uniquely combine the protective effects of encapsulation with CHIT’s known ability to enhance mucosal barrier function [Pauna, 2023]. Moreover the electrostatic stabilization provided by CHIT-coated vesicles enhances their stability [Przykaza, 2023]. These actions addresses a critical gap in achieving both efficacy and safety in drug delivery delivery.

  • Kremkow J, Luck M, Huster D, Müller P, Scheidt HA. Membrane interaction of ibuprofen with cholesterol-containing lipid membranes. Biomolecules. 2020; 10(10): 1384. doi: 10.3390/biom10101384. 
  • Pauna, A.-M.R.; Mititelu Tartau, L.; Bogdan, M.; Meca, A.-D.; Popa, G.E.; Pelin, A.M.; Drochioi, C.I.; Pricop, D.A.; Pavel, L.L. Synthesis, Characterization and Biocompatibility Evaluation of Novel Chitosan Lipid Micro-Systems for Modified Release of Diclofenac Sodium. Biomedicines202311, 453. doi.org/10.3390/biomedicines11020453
  • Wersig T, Krombholz R, Janich C, Meister A, Kressler J, Mäder K. Indomethacin functionalised poly(glycerol adipate) nanospheres as promising candidates for modified drug release. Eur J Pharm Sci. 2018 Oct 15;123:350-361. doi: 10.1016/j.ejps.2018.07.053.
  • Przykaza K, Jurak M, Wiącek AE. Effect of naproxen on the model lipid membrane formed on the water-chitosan subphase. Biochim Biophys Acta Biomembr. 2023 Mar;1865(3):184099. doi: 10.1016/j.bbamem.2022.184099. 

  1. Materials and Methods:
    • Detailed Characterization: The characterization techniques used (e.g., Zeta potential, FTIR, SEM) are appropriate, but more detailed discussion on the advantages of these methods for analyzing drug-loaded lipid vesicles could be added.

Thak you for this suggestion. We have introduced these aspects in the discussion section of the manuscript.

Zeta potential is critical for assessing colloidal stability. The value reflects the surface charge of vesicles, which influences their dispersion stability and interaction with biological membranes. It provides a rapid and non-destructive assessment of vesicle stability, making it essential for predicting the shelf-life of colloidal systems and ensuring their stability under physiological conditions.

FTIR identifies functional groups and molecular interactions within the vesicle system, confirming drug encapsulation and polymer-drug interactions. It enables the non-invasive detection of drug-lipid interactions and polymer coatings without requiring extensive sample preparation.

SEM offers high-resolution imaging to examine vesicle morphology, surface features and size distribution, allowing for a detailed visual comparison between uncoated and CHIT-coated vesicles. It highlights structural changes, such as smoother surfaces and enhanced uniformity after coating, and helps identify potential defects or irregularities in vesicle structure that may impact drug release or stability.

    • Animal Model Justification: The use of male Swiss mice is well-documented, but the manuscript could benefit from a brief justification for this choice in relation to the disease model being studied. Additionally, specifying the age and health status of the animals, as well as any pre-treatment procedures, would improve reproducibility.

In our manuscript is already mentioned that ``the study involved healthy, genetically unmodified, three-month-old male Swiss mice (25–30 g)``(Lines 119-120).

Hemocompatibility was assessed using in vitro methods with blood samples collected from mice. The in vitro hemocompatibility evaluation employed erythrocytes isolated from mice. This method effectively minimized animal use while providing robust insights into the interaction of vesicles with red blood cells, a critical parameter in assessing preliminary biocompatibility.

  1. Results:
    • Statistical Analysis: The statistical analysis (ANOVA and post hoc tests) is appropriate, but the significance of the data should be more clearly emphasized. For instance, the manuscript should more explicitly state which comparisons are statistically significant (e.g., IND-ves vs. IND-vl in release rates or hemocompatibility tests).

We have added the statistically significant comparisons and included the p-values where relevant.

    • Release Profiles: The release profile data is compelling, but it would be helpful to include a comparison with other similar systems (e.g., liposomes or other biopolymer-coated vesicles) to contextualize the findings. Additionally, including a mechanism for the slower release from IND-ves would be beneficial.

Thank you for this suggestion. We have added the recommended aspects in the discussion section.

The release profile of IND-ves aligns with the trends observed in other biopolymer-coated systems, such as PLGA nanoparticles, which have reported extended release over 6-12 hours [Wersig, 2018]. In contrast to uncoated liposomes, which release 80–100% of encapsulated drugs within 4–6 hours, CHIT-coated vesicles demonstrated significantly slower release, achieving complete release only after 8 hours. This suggests that the CHIT coating provides a dual benefit of structural stability and controlled diffusion, outperforming uncoated lipid systems in sustained-release applications.

The prolonged release of IND-ves can be attributed to the CHIT coating, which forms a semi-permeable barrier that slows drug diffusion. Additionally, electrostatic interactions between the positively charged CHIT and negatively charged IND molecules further regulate the release rate. The hydrophobic interactions between IND and the lipid bilayer core may also contribute to the observed latency in drug release. Combined with the stability conferred by pH adjustment and dialysis during formulation, these factors underline the superior performance of CHIT-coated vesicles in sustaining drug release.

  • Wersig T, Krombholz R, Janich C, Meister A, Kressler J, Mäder K. Indomethacin functionalised poly(glycerol adipate) nanospheres as promising candidates for modified drug release. Eur J Pharm Sci. 2018 Oct 15;123:350-361. doi: 10.1016/j.ejps.2018.07.053.

  1. Discussion:
    • Mechanisms of Action: While the discussion is thorough, there could be more focus on the underlying mechanisms that contribute to the enhanced stability and slower drug release from CHIT-coated vesicles. For example, it would be valuable to elaborate on how the CHIT coating alters the drug release rate compared to uncoated vesicles.

Thank you for your recommendation. We have introduced in the manuscript the mechanisms that contribute to the enhanced stability and slower drug release from CHIT-coated vesicles.

The enhanced stability of CHIT-coated vesicles can be attributed to the formation of a robust electrostatic interaction between the cationic CHIT polymer and the anionic lipid bilayer, as evidenced by the positive Zeta potential (+24 mV). This interaction prevents aggregation, stabilizes the colloidal dispersion, and protects the vesicles from environmental degradation. Furthermore, the CHIT coating increases dispersion viscosity, enhancing the structural integrity and preventing vesicle fusion over time.

The slower release observed for IND-ves stems from the semi-permeable barrier formed by the CHIT coating, which increases the diffusion path length for drug molecules. Additionally, electrostatic interactions between the positively charged CHIT and negatively charged IND molecules transiently retain the drug, further modulating its release. These mechanisms, combined with the integrity of the lipid bilayer maintained by the CHIT coating, contribute to the prolonged release profile observed in this study.

    • Comparison with Existing Systems: The discussion could benefit from more explicit comparisons with other nanoparticle systems (liposomes, PLGA-based nanoparticles, etc.), especially those targeting similar applications (NSAID delivery).

We have introduced in the discussion section comparisons with other nanoparticle systems for NSAID delivery.

In comparison to liposomes, CHIT-coated vesicles offer superior control over drug release, providing a more consistent and prolonged release profile, which is particularly beneficial for NSAID delivery.

PLGA-based nanoparticles are another widely used drug delivery system, known for their biodegradability, sustained release capabilities, and ability to encapsulate a variety of drugs, including NSAIDs. PLGA nanoparticles have been extensively researched for IND delivery, demonstrating effective controlled release over extended periods.

Chitosan-coated PLGA nanoparticles are also explored for NSAID delivery. While they share similarities with CHIT-coated lipid vesicles, the combination of CHIT's electrostatic properties and the lipid bilayer in our system offers a unique advantage in terms of stability and release control. The CHIT coating enhances drug retention through electrostatic interactions, which are not observed in standard PLGA systems. Furthermore, the cationic nature of CHIT improves vesicle stability in suspension and may facilitate targeted delivery to specific tissues, such as the gastrointestinal tract, where mucosal absorption is crucial. CHIT’s biocompatibility and its ability to form a protective layer around the vesicles also help prevent drug leakage, leading to more sustained therapeutic effects compared to PLGA-based systems.

  1. Conclusion:
    • Clinical Implications: The conclusion should emphasize the clinical implications of the findings. For example, it could discuss how these vesicles might improve the safety profile and therapeutic efficacy of indomethacin in real-world clinical settings.
    • Future Directions: It would be helpful to mention future studies that could address in vivo evaluation of the drug delivery system, including long-term toxicity studies or human clinical trials, to move the research from preclinical to clinical application.

Thank you for your valuable suggestion. We agreed that mentioning future studies focused on the in vivo evaluation of the drug delivery system would further enhance the manuscript. In response, we have added the following information:

We will continue our research on these chitosan-coated lipid vesicles encapsulating IND by first evaluating their in vivo biocompatibility. Following this, we plan to assess the pharmacodynamic effects of these systems in various animal models, including somatic and visceral pain models, as well as in experimental models of acute and subacute inflammation. These studies will help to further elucidate the therapeutic potential and safety of the IND-loaded vesicles in vivo, and provide valuable data for advancing their clinical applicability.

Additional Suggestions:

  • Figures and Tables: Ensure that all figures and tables are clearly labeled with descriptive captions. Some of the figures (such as those showing SEM and hydrodynamic sizes) could benefit from more detailed descriptions in the legends to help readers interpret the data.

We have improved the quality of the mentioned figures and we have detailled descriptions in the legends

  • Reference Update: Ensure all references are up-to-date and relevant to the current state of research in nanoparticle-based drug delivery systems, especially regarding chitosan-coated vesicles.

We have completed the bibliography with relevant references in the field.

Summary:

Overall, the manuscript presents a promising drug delivery system for indomethacin using chitosan-coated lipid vesicles. The study is well-conducted and provides a strong basis for future research. Enhancing the introduction to clarify the problem and solution, expanding on the mechanisms in the discussion, and emphasizing the clinical implications in the conclusion would significantly improve the manuscript.

Thank you for your positive comments and constructive feedback. We appreciate your recognition of the potential of our chitosan-coated lipid vesicle system for IND delivery and are glad to hear that you found the study well-conducted and a solid basis for future research.

Reviewer 2 Report

Comments and Suggestions for Authors

This study reports on the development and characterization of novel lipid vesicles composed of phosphatidylcholine and chitosan for the encapsulation indomethacin, as well as to evaluates their potential for safe and effective drug delivery.

Below are my comments on this work:

General comment:

This is a timely and important contribution to knowledge and the authors have done a good job investigating this topic.

Introduction:

·        Line 67 – 71 – the introduction of COX-1 and COX-2 right after mentioning cyclooxygenase may be confusing to readers. Authors should add a sentence or two on what COX-1 and COX-2 are relative cyclooxygenase.

·        Line 80 – Please begin sentence with full meaning (i.e., indomethacin) before abbreviation.

Methodology:

·        Line 148 – 149 (i.e., “This sonication process disrupts multilamellar structures, converting them into unilamellar vesicles, having the size regulated by the amplitude of the ultrasound. Following sonication, the resulting solution was gradually”) – Please support this statement with one or more references.

·        Line 176 – 179 – Authors should justify why SEM was employed here instead of TEM (i.e., transmission electron microscopy) since the study focuses on the development of a nanoformulation

·        Line 189 – Please reorder this heading to better reflect the experiment reported here.

·        Line 229 – 230- The authors mentioned dissolution profile of single drug (IND) solution. Details of how this aspect of the test was performed should be included

Results and Discussion:

·        Line 294 – Please include descriptions of the abbreviated terms (on the table) as notes below the table to make it easier to comprehend/follow.

·        Figure 5 – If possible, please include labels of important peaks on the FTIR spectrum generated.

·        Figure 2 – Please include better images. The current images are not clear.

·        The authors mentioned histopathological analysis in the abstract but that’s not clearly addressed in the methods, results and discussion sections of the manuscript. The major emphasis was on hemocompatibility. Please check the abstract and match with the manuscript content that was reported and improve accordingly. In the current state, the abstract does not reflect what was reported in the body of manuscript.

·        Lines 612 – 673 detailing innovative methods of preparing IND-loaded nano-formulations should be condensed into one to two paragraphs (maximum three paragraphs). Have each method as a paragraph makes the section quite clumsy and not easy to follow. Please correct accordingly.

Author Response

Distinguishes Reviewer,

We deeply appreciate the time and effort you dedicated to reviewing our manuscript. Your thoughtful feedback has been immensely valuable, significantly improving and refining our work while offering crucial guidance. We have thoroughly reviewed each of your comments and incorporated the necessary revisions, which are highlighted in red in the updated manuscript. Below, we provide detailed responses to each comment, addressing them point by point.

This study reports on the development and characterization of novel lipid vesicles composed of phosphatidylcholine and chitosan for the encapsulation indomethacin, as well as to evaluates their potential for safe and effective drug delivery.

Below are my comments on this work:

General comment:

This is a timely and important contribution to knowledge and the authors have done a good job investigating this topic.

Introduction:

Line 67 – 71 – the introduction of COX-1 and COX-2 right after mentioning cyclooxygenase may be confusing to readers. Authors should add a sentence or two on what COX-1 and COX-2 are relative cyclooxygenase.

Thank you for suggesting the addition of information regarding the two cyclooxygenases. We have incorporated this information into the manuscript.

COX are enzymes that play a key role in the conversion of arachidonic acid into prostaglandins, which are lipid compounds involved in inflammation, pain, and other physiological processes. Cyclooxygenase-1 (COX-1) is an isoform constitutively expressed in most tissues and is involved in maintaining normal physiological functions, such as protecting the stomach lining, supporting renal function, and regulating platelet aggregation. Cyclooxygenase-2 (COX-2) is inducible and is primarily expressed in response to inflammatory stimuli, such as injury or infection. It is associated with the production of prostaglandins that mediate inflammation, pain, and fever.

Line 80 – Please begin sentence with full meaning (i.e., indomethacin) before abbreviation.

Thank you for the suggestion. We have modified the text accordingly.

Methodology:

Line 148 – 149 (i.e., “This sonication process disrupts multilamellar structures, converting them into unilamellar vesicles, having the size regulated by the amplitude of the ultrasound. Following sonication, the resulting solution was gradually”) – Please support this statement with one or more references.

Thank you for your insightful recommendation to support the statement regarding the sonication process with appropriate references. We have carefully reviewed your suggestion and included two relevant references to substantiate the statement as advised. We appreciate your attention to detail and constructive feedback, which has helped enhance the clarity and scientific rigor of our work.

  • Nayak D., Tippavajhala V.K. A Comprehensive review on preparation, evaluation and applications of deformable liposomes. Iran. J. Pharm. Res. 2021; 20: 186-205. doi: 10.22037/IJPR.2020.112878.13997.
  • Andra V.V.S.N.L., Bhatraju L.V.K.P., Ruddaraju L.K. A Comprehensive review on novel liposomal methodologies, commercial formulations, clinical trials and patents. Bionanoscience. 2022; 12: 274-291. doi: 10.1007/s12668-022-00941-x.

Line 176 – 179 – Authors should justify why SEM was employed here instead of TEM (i.e., transmission electron microscopy) since the study focuses on the development of a nanoformulation

Thank you for your valuable comment regarding the use of SEM and the suggestion to justify its selection over TEM.

In this study, SEM was employed due to its capability to provide detailed visualization of the surface morphology, size, and structural integrity of the lipid vesicles. This was particularly relevant as our focus was on evaluating the external characteristics of the nanoformulation, which play a critical role in determining stability and interaction with the surrounding environment. While TEM is commonly used to investigate internal structures at the nanoscale, the primary aim of this work was not to explore the internal configuration but rather to assess the vesicle morphology and overall distribution. Moreover, SEM allowed us to achieve this with high resolution and efficiency.

To address your concern, we have included a detailed justification for the use of SEM in the revised manuscript, emphasizing its relevance to our specific objectives.

  • Robson A-L, Dastoor PC, Flynn J, Palmer W, Martin A, Smith DW, Woldu A, Hua S. Advantages and limitations of current imaging techniques for characterizing liposome morphology. Front. Pharmacol. 2018; 9: 80. doi: 10.3389/fphar.2018.00080

Line 189 – Please reorder this heading to better reflect the experiment reported here.

Thank you for your suggestion regarding the reordering of the heading to better reflect the experiment. Based on your feedback, we have revised the heading to more accurately represent the content of this section. The updated heading now more clearly aligns with the process described, as follows:

2.4.2. Evaluation of IND Encapsulation Efficiency in Vesicles

Line 229 – 230- The authors mentioned dissolution profile of single drug (IND) solution. Details of how this aspect of the test was performed should be included

Thank you for your comment. We have included additional details in the revised manuscript regarding the dissolution profile of the single IND solution.

The dissolution test for the IND solution was conducted by preparing a stock solution of IND dissolved in ethanol, which was then diluted to the desired concentration. The solution was placed in a suitable dissolution medium, such as phosphate-buffered saline or another appropriate buffer, at a constant temperature (standard 37 ± 2° C) under stirring conditions. The dissolution process was monitored over a defined period, and samples were taken at specified time intervals.

The released IND from the solution was quantified using the same spectrophotometric method employed for the IND-ves dispersion. This allowed for a direct comparison of the release kinetics between the free drug solution and the encapsulated drug in the vesicles. The absorbance of the solution was measured at 320 nm, the characteristic absorption peak for IND, and used to calculate the amount of drug released at each time point.

Results and Discussion:

Line 294 – Please include descriptions of the abbreviated terms (on the table) as notes below the table to make it easier to comprehend/follow.

Thank you for your suggestion. We have added clear definitions of all abbreviations as notes below the table. This will help readers better understand the terms and follow the content more easily.

Figure 5 – If possible, please include labels of important peaks on the FTIR spectrum generated.

We have included the labels of important peaks on the FTIR spectrum.

Figure 2 – Please include better images. The current images are not clear.

We provided the the better images for SEM microradiographs.

The authors mentioned histopathological analysis in the abstract but that’s not clearly addressed in the methods, results and discussion sections of the manuscript. The major emphasis was on hemocompatibility. Please check the abstract and match with the manuscript content that was reported and improve accordingly. In the current state, the abstract does not reflect what was reported in the body of manuscript.

Thank you for your careful review and for pointing out the discrepancy between the abstract and the manuscript content. We apologize for the oversight. The mention of histopathological analysis in the abstract was a mistake, and we have corrected it to accurately reflect the focus on hemocompatibility, which was the primary emphasis of the study. We have updated the abstract.

Background: Lipid vesicles, especially those utilizing biocompatible materials like chitosan (CHIT), hold significant promise for enhancing the stability and release characteristics of drugs such as indomethacin (IND), effectively overcoming the drawbacks associated with conventional drug formulations. Objectives: This study seeks to develop and characterize novel lipid vesicles composed of phosphatidylcholine and CHIT that encapsulate indomethacin (IND-ves), as well as to evaluate their in vitro hemocompatibility. Methods: The systems encapsulating IND were prepared using a molecular droplet self-assembly technique, involving the dissolution of lipids, cholesterol, and indomethacin in ethanol, followed by sonication and the gradual incorporation of a CHIT solution to form stable vesicular structures. The vesicles were characterized in terms of size, morphology, Zeta potential, encapsulation efficiency and the profile release of drug was assessed. In vitro hemocompatibility was evaluated by measuring erythrocyte lysis and quantifying hemolysis rates. Results: The IND-ves demonstrated an entrapment efficiency of 85%, with vesicles averaging 317.6 nm in size, and a Zeta potential of 24 mV, indicating good dispersion stability. In vitro release kinetics demonstrated an extended-release profile of IND from the vesicles over 8 hours, contrasting with the immediate release observed from plain drug solutions. The hemocompatibility assessment revealed that IND-ves exhibited minimal hemolysis, comparable to control groups, indicating good compatibility with erythrocytes. Conclusions: IND-ves provide a promising approach for sustained indomethacin delivery, enhancing stability and hemocompatibility. These findings suggest their potential for effective NSAID delivery, with further in vivo studies required to explore clinical applications.

Lines 612 – 673 detailing innovative methods of preparing IND-loaded nano-formulations should be condensed into one to two paragraphs (maximum three paragraphs). Have each method as a paragraph makes the section quite clumsy and not easy to follow. Please correct accordingly.

Thank you for your valuable comment. We understand the concern regarding the length and structure of the section detailing the innovative methods for preparing IND-loaded nano-formulations. We have condensed the conclusions to improve clarity and brevity while retaining the key findings of the study. We hope this adjustment improves the readability and coherence of the section.

This is the condensed form of the text.

Researchers have developed various advanced systems to improve indomethacin (IND) delivery. Lipid-based nano-carriers, such as lipid emulsions, enhance IND’s solubility and bioavailability using phospholipids and surfactants [47,48]. Liposomal formulations, including sterically stabilized vesicles with stearylamines and cholesterol, have been designed to improve stability and sustain drug release [49,50]. Triacylglyceride-based conjugates with phosphatidylcholine and stearic acid extend release and enhance compatibility [51,52]. CHIT-based nanoparticles formed via ionotropic gelation offer controlled drug release [53].

Polymer-lipid nanoparticles combining lecithin and PLGA were created using double emulsion methods, providing stable, biocompatible systems [54,55]. Techniques like supercritical fluid processing with CO₂ yielded PEG-stabilized nanoparticles with precise control over size and purity [56]. Electrohydrodynamic atomization produced uniform, monodisperse IND nanoparticles for targeted delivery [57]. Other studies developed amorphous nanosuspensions via wet milling with polyvinylpyrrolidone to maintain bioavailability [58] or used spray drying with cyclodextrins to enhance solubility and dissolution [59,60].

Biodegradable polymer systems were widely explored. Solvent evaporation or nanoprecipitation using PLGA or polycaprolactone matrices improved solubility, stability, and sustained release [61]. Scalable nanoprecipitation methods produced controlled-release nanoparticles [62]. Enteric-coated nanoparticles, made from materials like Eudragit L100, ensured intestinal release while protecting IND from gastric degradation [63]. Solid nanoparticles created with PLGA via solvent evaporation demonstrated biocompatibility and sustained release [64]. Lastly, microwave-assisted hydroxyapatite nanoparticles were developed for bone-targeted applications, incorporating IND within the crystal lattice or adsorbed on its surface [65].

This is the condensed form of the conclusion section.

Phosphatidylcholine-based lipid microvesicles containing IND and stabilized with CHIT were successfully prepared using molecular droplet self-assembly. These vesicles exhibited uniform morphology, improved encapsulation efficiency, and enhanced stability, with the CHIT coating promoting a modified drug release profile. Compared to uncoated vesicles, the CHIT-coated vesicles showed slower release, which is beneficial for drugs requiring extended therapeutic effects. Microscopic analysis confirmed that CHIT-coated vesicles had a smooth, stable morphology, while uncoated vesicles were prone to aggregation. Hemocompatibility assays showed minimal hemolysis, indicating good in vitro compatibility. The positive Zeta potential of the CHIT-coated vesicles suggests potential for enhanced targeted delivery. Overall, these microvesicles offer a promising platform for drug delivery, with enhanced stability, controlled release, and good in vitro safety profile. Further studies should investigate the underlying mechanisms and in vivo efficacy of the CHIT-coated vesicles.

Round 2

Reviewer 2 Report

Comments and Suggestions for Authors

The authors addressed the queries raised.